# DeepPCM: Predicting Protein-Ligand Binding using Unsupervised Learned Representations

## Abstract

*In-silico* protein-ligand binding prediction is an ongoing area of research in computational chemistry and machine learning based drug discovery, as an accurate predictive model could greatly reduce the time and resources necessary for the detection and prioritization of possible drug candidates. Proteochemometric modeling (PCM) attempts to make an accurate model of the protein-ligand interaction space by combining explicit protein and ligand descriptors. This requires the creation of information-rich, uniform and computer interpretable representations of proteins and ligands. Previous work in PCM modeling relies on pre-defined, handcrafted feature extraction methods, and many methods use protein descriptors that require alignment or are otherwise specific to a particular group of related proteins. However, recent advances in representation learning have shown that unsupervised machine learning can be used to generate embeddings which outperform complex, human-engineered representations. We apply this reasoning to propose a novel proteochemometric modeling methodology which, for the first time, uses embeddings generated via unsupervised representation learning for both the protein and ligand descriptors. We evaluate performance on various splits of a benchmark dataset, including a challenging split that tests the model's ability to generalize to proteins for which bioactivity data is greatly limited, and we find that our method consistently outperforms state-of-the-art methods.

## 1 Introduction

A main goal of cheminformatics in the area of drug discovery is to model the interaction of small molecules with proteins *in-silico*. The ability to accurately predict the binding affinity of a ligand towards a biological target without the need to conduct expensive *in-vitro* experiments has the potential to accelerate the drug development process by enabling early prioritization of promising drug candidates (Cortés-Ciriano et al., 2015). A common approach is to train a machine learning algorithm to predict the binding affinity of ligands towards a certain biological target using a training set of compounds that have been experimentally measured on this target. This modality is commonly referred to as a quantitative structure-activity-relationship (QSAR) model (van Westen et al., 2011).

QSAR models can be broadly classified into two types: single-task QSAR models and multi-task QSAR models (Figure 1). In single-task QSAR modeling, a model is trained separately for each protein to predict a binary or continuous outcome (binding vs not-binding or the binding affinity) given a compound input. The machine learning model used could be anything from logistic regression to deep neural networks (Lenselink et al., 2017; Cherkasov et al., 2014).In multi-task modeling, a single model is trained to predict binding across multiple proteins simultaneously, allowing the model to take advantage of the correlations in binding activity between compounds on different targets (Caruana, 1997; Yuan et al., 2016). This is done, for example, by using a neural network with multiple output nodes where each output node corresponds to a different protein. Thus, multiple outputs are predicted given a compound input (Simões et al., 2018; Dahl et al., 2014).

While these methods have been employed on various protein targets, there are methodological concerns to their use (Lima et al., 2016; Mitchell, 2014). Both single-task and multi-task models must be retrained from scratch if one wishes to incorporate binding data for a new protein, and both cannot be used at all to make predictions on new protein targets for which experimental data is absent (van Westen et al., 2011).

An attractive solution to this problem is to also include protein information in a so called proteochemometric model (Figure 1). The additional protein information enables a PCM model to directly utilize similarities between proteins for bioactivity modeling (Cortés-Ciriano et al., 2015; van Westen et al., 2011). This leads to many potential benefits over classical QSAR single-task and multi-task modeling. First, as proteins are explicitly represented by a defined featurization, PCM models can be used to make activity predictions on proteins without pre-existing bioactivity data, which is impossible with single-task or multi-task modeling. They can also be used to model binding on proteins for which experimental data may be too limited to train an effective single-task model (van Westen et al., 2011). Second, with an expressive protein descriptor, a model could leverage similarities and differences between proteins directly to model their binding behaviors, rather than merely using correlations found among the compounds they bind to, as in multi-task modeling, or ignoring protein relationships altogether as in single-task modeling (Cortés-Ciriano et al., 2015; van Westen et al., 2011). Consequently, PCM models have found success on a variety of protein targets, and using many different machine learning methods, including random forests and SVMs (Ballester & Mitchell, 2010; Weill et al., 2011; van Westen et al., 2013; Shiraishi et al., 2013; Cheng et al., 2012).

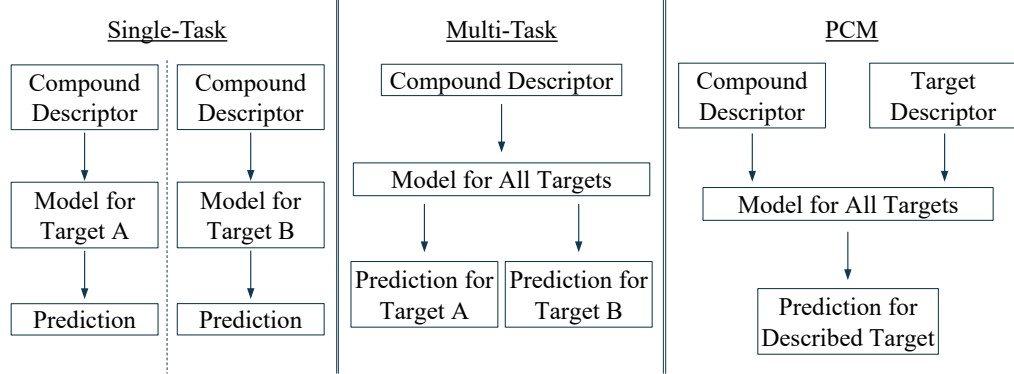

Figure 1: Depiction of the architectures of different ligand-target binding-affinity prediction models.

Recent advances in the field of Deep Learning have also led to its use in QSAR single and multi-task modeling as well as PCM modeling (Lenselink et al., 2017; Menden et al., 2013; LeCun et al., 2015; Schmidhuber, 2015). Lenselink et al. (2017) compared deep learning methods against other machine learning methods for PCM, single-task QSAR, and multi-task QSAR models on a benchmark dataset, and the authors found that PCM models using deep neural networks outperform other machine learning methods as well as single-task and multi-task QSAR models.

All of the aforementioned works make features for both the small molecules and the proteins based on hand-crafted feature extraction protocols. Small molecules are often represented by counting and aggregating smaller substructures, while proteins are often represented by aggregation of computed physio-chemical features of their amino-acids or by encoding the differences in aligned sequences.

In many application domains of Deep Learning, recent research has shown that these methods generally work better when the input data representation is lower-level and unabstracted, allowing the model to learn hierarchical features directly rather than relying on features which are hand-crafted by humans (LeCun et al., 2015; Schmidhuber, 2015). For example, this is the case in computer vision, where deep learning on pixel value features has been the state of the art for several years (Schmidhuber, 2015). In cases where the input space is too high-dimensional, and especially if there is not enough labeled data to train a model end-to-end, unsupervised representation learning is used to generate lower-dimensional embeddings, again relying on machine learning, rather than human engineering, for feature extraction. This is the case in natural language processing, as well as in video analysis (Srivastava et al., 2015; Erhan et al., 2010; Mikolov et al., 2013).

In this work, we follow this reasoning and utilize such unsupervised-learned embeddings to represent both the ligand and protein spaces for proteochemometric modelling. To the best of our knowl-

edge, we combine, for the first time, compound and protein representations both generated by deep learning models that were pretrained on an unsupervised learning task, as inputs to a PCM model. Moreover, we believe that this is the first work which combines unsupervised-learned embeddings of biological and chemical entities simultaneously in order to solve a downstream task.

## 1.1 COMPOUND DESCRIPTORS

The following section describes the currently used handcrafted ligand descriptors, evaluates their properties, and introduces the ligand descriptors that we use for our model — known as CDDD descriptors — which are generated via unsupervised machine learning.

**Handcrafted Compound Descriptors**   The state-of-the-art compound descriptors that are used for a vast majority of PCM and other chemoinformatics tasks are different varieties of circular fingerprints. In binary keyed circular fingerprints, each bit refers to a specific substructure's presence or absence in the molecule, while in counts format the bits refer to the number of occurrences of the substructure (van Westen et al., 2011; Glen et al., 2006; Rogers & Hahn, 2010). As the number of potential substructures is vast ($\sim 2^{32}$), the resulting sparse set of bits is usually hashed and folded to a much smaller size ($\sim 10^3$) at the expense of hash and bit collisions (Rogers & Hahn, 2010). These structure-based descriptors can be augmented with physicochemical descriptors, such as DRAGON or PaDEL, along with other chemical descriptors like atom identity/type, MACCs keys, and topological indices (Mauri et al., 2006; Yap, 2011).

**Issues**   Circular fingerprints contain information about only the presence or absence of certain substructures in the compound – they thus fail to capture the shape or arrangement of those substructures within the compound. Furthermore, fingerprints rely on a hashing protocol to compress the millions of different substructures that are recorded into a smaller vector – as a result, hash collisions can mean that completely different substructures can correspond to the same fingerprint bit. This also means that one cannot determine which exact substructures are responsible for the model prediction.

**CDDD – Ligand Descriptor**   To avoid these issues, we use the CDDD (Continuous and Data Driven Molecular Descriptors), a model for the generation of lower-dimensional representation vectors of molecules developed by Winter et al. (2019). This model uses a recurrent autoencoder trained on the task of translating non-canonical SMILES string representations of compounds into their canonical form. After unsupervised learning on approximately 72 million compound SMILES, a given compound is represented by the 512-length bottleneck of the translation model, and thus is encoded into a 512-length vector. These embeddings have been shown to be effective on QSAR prediction and virtual screening tasks (Winter et al., 2019).

CDDD descriptors offer a unique, compact, and continuous vector representation for each compound, as opposed to fingerprints, which are non-unique, discrete, and must be hashed to be made compact. Molecules with the same substructures but differently arranged will correspond to different vectors. Additionally, these unsupervised-learned descriptors have demonstrated competitive or superior results compared to molecular fingerprints on a variety of other tasks, indicating their ability to effectively represent compound properties and behaviors (Winter et al., 2019). A diagram can be found in Appendix section D.

## 1.2 PROTEIN DESCRIPTORS

Next, we describe the various methodologies used for handcrafted protein descriptors. There are many different examples of highly specialized protein feature-extraction protocols used for PCM models that operate over a narrow range of targets. We describe the broad categories, examine their drawbacks and introduce the unsupervised protein descriptor which we use, known as UniRep.

**Handcrafted Protein Descriptors**   Most commonly used protein descriptors can be described as either amino-acid-based, or sequence-based. Amino-acid-based descriptors are computed using some combination of physicochemical and structural properties of individual amino acids. For example, the commonly used Z-scale descriptors and their counterpart ProtFP descriptors are constructed by computing various amino acid physicochemical properties (such as hydrophobicity or polarity), taking a PCA over these properties, and then representing each amino acid by its first few principal components (Sandberg et al., 1998; van Westen et al., 2013). These descriptors can be combined with structural information descriptors, such as normalized van der Waals volume, charge,

and the secondary structure of the protein at that amino acid residue (Lapins et al., 2013). The whole protein is then represented by aggregating the amino-acid descriptors over the whole sequence.

Sequence-based descriptors are constructed by taking aligned sequences or regions and using encodings to represent the amino acid identity at a given location in the aligned region. For example, in Nabu et al. (2015), the authors build a model that predicts the bioactivity on mutated vs wild type penicillin binding proteins based on the positions of the mutations in the sequence. This was done by aligning the protein sequences and uniquely representing the positions a mutation may have occurred at with a one-hot-encoding. Thus, from 75 potential mutation sites, a 75-length vector is returned for each protein. Other variations of this technique use motifs or encodings of multiple varying segments (Lapinsh et al., 2005).

Additionally, on proteins for which 3D-information is available, this information can be incorporated into the protein descriptor. Methods vary – in general, 3D-models, for example crystal structures or water-field maps, are first aligned. Subsequently, descriptors are built based on amino acid identity at specific positions at the binding pocket or taking a PCA over the aligned field maps (van Westen et al., 2013; Lapinsh et al., 2005; Subramanian et al., 2016; Kruger & Overington, 2012).

**Issues** Amino acid-based descriptors require crude aggregation or binning methods, as they must convert variable length sequences of amino acids into a constant-length vector. For example, a common strategy is to divide the sequence into equal-length segments, then average over the descriptors for each segment (Lenselink et al., 2017; van Westen et al., 2013). This leads to the undesirable property where a single amino acid insertion early in the sequence would shift all subsequent segments, thus changing every descriptor bit despite a small change to the protein itself. Additionally, motifs and functional domains have variable numbers of amino acids which are not necessarily contiguous, so averaging over fixed-length contiguous segments can fail to capture relevant features consistently.

Sequence and 3D-structure based descriptors must be aligned, which restricts the usable bioactivity data to only a very small fraction of closely related proteins for which an alignment is meaningful. These descriptors also cannot be applied to new regions of the protein space, where bioactivity measurements of similar proteins are unavailable. Essentially, these methods build descriptors that are explicitly based on the differences between a few proteins, which greatly limits the scope of the problems that can be approached with such descriptors, as well as the amount of data that can be leveraged to train models. These data availability issues are exacerbated when using 3D-descriptors, since 3D structures are only available for a subset of proteins.

**UniRep – Protein Descriptor** We use UniRep developed by Alley et al. (2019), which uses a multiplicative LSTM architecture on approximately 24 million protein amino acid sequences taken from UniRef50 (Krause et al., 2016). The UniRep model is trained on the next-character-prediction task, where the LSTM predicts the next amino acid in the protein sequence given the previous amino acids. To generate fixed-length embeddings, the hidden states of the model that are generated during the forward pass are averaged over the sequence dimension. For implementation details, refer to Alley et al. (2019). The pre-trained model can be used out-of-the-box, taking as input amino acid sequences and outputting embeddings of length 64, 256, or 1900 depending on the architecture used. For the PCM model which we propose here, we found that the 256-length embeddings performed best, and so we will use UniRep to refer to the 256-length UniRep descriptor in this paper. A diagram can be found in Appendix section E

UniRep descriptors offer many advantages over their handcrafted counterparts. The descriptors are alignment-free sequence-based representations, which therefore allow the utilization of protein bioactivity assay data across families and species instead of just highly similar proteins, greatly increasing the available training data. Moreover, these descriptors do not require 3D-structure information, which is available for only a small subset of proteins. Additionally, there is no need for binning or averaging across arbitrary length contiguous chunks of sequence.

## 2 METHODS

### 2.1 DATASET AND EVALUATION

We evaluate on a large-scale benchmark PCM dataset created in Lenselink et al. (2017), which contains 310k compound-protein bioactivity measurements taken from exclusively the highest-confidence bioactivity assay data in ChEMBL. The dataset is comprised of 1226 unique human proteins from a range of protein families, and 190k unique compounds.

For validation purposes, we use three different types of hold-out-sets: random and temporal splits, which are included in Lenselink et al. (2017), and low-coverage-protein splits, which is a new and more challenging criterion/splitting. Random splits randomly divide the bioactivity measurements into train, valid, and test sets, with the validation set used for early stopping. A drawback of the random split is that it can assign bioactivity measurements from the same experimental assay into the training and test sets. Since a single assay can involve similar, congeneric compounds, it is likely that there will be compounds in the training set that are highly similar to compounds in the test set and measured on the same protein. This split can thus report overly optimistic results (Lenselink et al., 2017). Therefore, experiments are performed on a temporal split, which attempts to overcome this problem by splitting measurements from experimental batches done during or after 2013 into the test set, and using batches done pre-2013 for the training and validation set.

Ultimately, we would like to assess the ability of the model to generalize to protein targets that only have very few experimental measurements for training. This setting is more challenging but also more relevant for demonstrating a PCM model's potential, since PCM models should be able to utilize relationships in the protein descriptor space to infer on proteins that have not been well covered in the training set. We therefore apply a final, more challenging split – the low-coverage split. Here, we create several data folds by randomly selecting small numbers of proteins and then randomly selecting 90% of the bioactivity measurements on these proteins to be held out as a test set. The remaining 10% of the measurements on these proteins are included in the training set, along with all the bioactivity measurements from the other proteins. Thus, the hold-out set of the low-coverage split only contains proteins for which there is low data coverage in the training set.

All hyperparameter optimization of our model was performed on the temporal split, and the best performing hyperparameter scheme on the validation set of the temporal split was used to train models on the random and low-coverage-protein splits. Early stopping on the validation set was used for the random and temporal splits, while the low-coverage splits were trained for a fixed number of epochs. For the random and temporal splits, we use bootstrapping over 25 bootstrapped samples to estimate the standard error of the various models, and we compute Wilcoxon signed-rank-test p-values over the bootstrapped samples to determine the significance of our results, while on the low-coverage split we compute Wilcoxon p-values over the paired held-out proteins (Demšar, 2006).

### 2.2 DESCRIPTORS

For our models, we use the unsupervised-learned UniRep and CDDD descriptors as described above. The handcrafted benchmark model which we compare against uses 4096-bit circular fingerprint descriptors plus six computed molecular physicochemical properties for the ligand. For the protein, the handcrafted benchmark computes amino-acid-property-based features and averages these properties across 20 equal-length parts of the protein sequence (Lenselink et al., 2017).

### 2.3 MODELS

**DeepPCM** The DeepPCM model is a feedforward neural network with fully connected layers. We first apply $\mathcal{N}(0, 0.01)$ Gaussian noise to the compound inputs and $\mathcal{N}(0, 0.05)$ Gaussian noise to the protein inputs for data augmentation and generalization purposes. The noised inputs are passed through separate bottleneck layers, each with 128 nodes. The bottleneck outputs and original noised inputs are concatenated together, to allow the rest of the model to use either the transformed features from the separate bottlenecks, or to learn features from the raw compound and protein inputs – a trick used in many recent neural network architectures (Huang et al., 2017; He et al., 2016). This concatenated vector is fed into two fully connected layers, with 2048 and 1024 nodes

Table 1: Results for the random and temporal split experiments performed on the benchmark dataset. Results for the Single- and Multi-Task QSAR model (*) were taken from Lenselink et al. (2017). Bold indicates results where our model exhibited highly statistically significant ($p < 0.0005$) improvement over the Benchmark model. NIB:No-Interaction-Terms Baseline; HD:Handcrafted Descriptors; UD: Unsupervised-learned Descriptors; UP: Unsupervised Protein Descriptor; UC: Unsupervised Compound Descriptor; HP: Handcrafted Protein Descriptor; HC: Handcrafted Compound Descriptor

| Model | Random | | Temporal | |
|---|---|---|---|---|
| | MCC | BEDROC | MCC | BEDROC |
| Single-Task QSAR * | 0.53±0.07 | 0.91±0.05 | 0.22±0.08 | 0.73±0.06 |
| Multi-Task QSAR * | 0.57±0.07 | 0.92±0.05 | 0.26±0.07 | 0.76±0.06 |
| NIB + HD | 0.529±0.012 | 0.954±0.004 | 0.287±0.016 | 0.776±0.012 |
| NIB + UD | 0.543±0.006 | 0.954±0.003 | 0.304±0.005 | 0.787±0.006 |
| Benchmark | 0.573±0.097 | 0.958±0.012 | 0.304±0.054 | 0.799±0.020 |
| DeepPCM + UP + HC | 0.619±0.004 | 0.968±0.002 | 0.317±0.007 | 0.820±0.010 |
| DeepPCM + HP + UC | 0.589±0.031 | 0.969±0.005 | 0.351±0.013 | 0.828±0.015 |
| **DeepPCM + UP + UC** | **0.630±0.008** | **0.973±0.003** | **0.353±0.007** | **0.839±0.007** |

respectively. Dropout regularization is used throughout. Details about the model architecture and hyperparameters can be found in the Appendix sections A and F.

**No-Interaction-Terms Baseline** In addition, to evaluate the contribution from compound-protein interaction terms, we include a no-interaction-terms model that does not allow information flow between proteins and compounds. This model processes protein and compound inputs in separate networks, with the final predictions from each network averaged to then give a final output for the model. A diagram is included in the Appendix section B.

**Benchmark Model** We compare against a PCM model developed in Lenselink et al. (2017), which uses a three-layer fully-connected pyramidal neural network. A diagram can be found in the Appendix section C. The authors compare this model against single-task and multi-task QSAR models as well as other deep learning architectures on their benchmark PCM dataset and find this model to perform best.

## 2.4 METRICS

The models and parameter settings are compared using Matthews Correlation Coefficient (MCC) and Boltzmann-Enhanced ROC (BEDROC) (Lenselink et al., 2017). The MCC score represents the overall model quality and is especially useful for measuring performance on unbalanced datasets. The BEDROC score is a metric that represents the effectiveness of the model for compound prioritization since often, only a small subset of *in-silico* screened compounds can be tested experimentally, a useful model will rank active compounds very highly (Truchon & Bayly, 2007). The BEDROC score represents this by weighting the ROC results such that 80% of the BEDROC score comes from the top 8% of predicted actives. Thus, it is analogous to a ROC-50 score.

## 3 RESULTS

Overall, on all splits and metrics, we find that our method outperforms the benchmark model with high statistical significance ($p < 0.0005$). On the random split, we find a 10% improvement in MCC from using our model compared to the benchmark, and a 10% improvement in MCC compared to the best multi-task QSAR model run by Lenselink et al. (2017) on this dataset. On the temporal split, the improvement is greater: 16% increase in MCC compared to the benchmark and 36% increase in MCC compared to the best multi-task model.

On the low-coverage-split, we also find that our method significantly outperforms the benchmark. In Figure 2a, the per-protein-target MCC results of our model vs the benchmark model are plotted. Our model, based on the unsupervised-learned descriptors, shows on average better performance.

When applying a Wilcoxon signed-ranked test we find this difference to be highly significant ($p = 4.1*10^{-7}$).

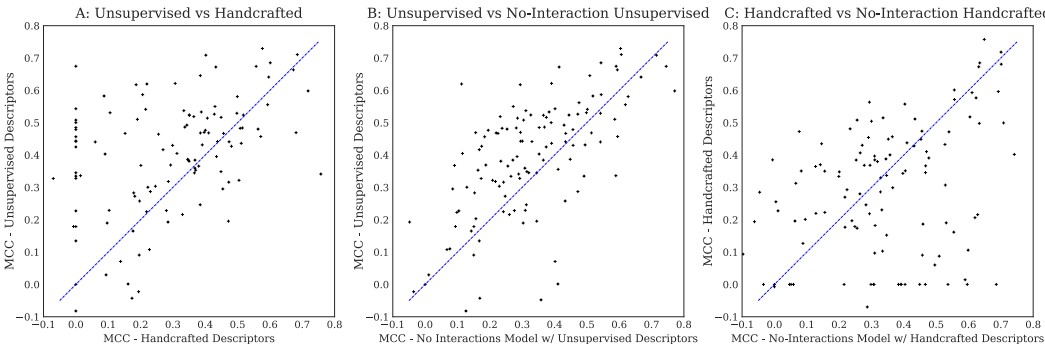

Figure 2: Scatterplot of pairwise MCC performance per protein on the low-coverage split. Panel A compares the DeepPCM model with unsupervised-learned descriptors to the benchmark model with handcrafted descriptors. Panels B and C compare the performance of the full models versus the No-Interaction-Terms models when using unsupervised-learned or handcrafted descriptors respectively. The blue line indicates the diagonal, where performance for both models would be equal.

## 4 DISCUSSION

The results are consistent with our expectation that the descriptors generated via unsupervised representation learning would be significantly more powerful than the handcrafted protein and compound descriptors.

In order to elucidate the individual impact of each descriptor type we also train models using unsupervised descriptors for either the ligand or the protein, and handcrafted descriptors for the other. We find that replacing the handcrafted compound descriptor with the unsupervised compound descriptor is responsible for the largest increase in performance on the temporal split, with the unsupervised protein descriptor providing a very small improvement (see Table 1). We note that we did not tune the network architecture or hyperparameters for these descriptor combinations  as a result, we might expect performance to be slightly different if the models were to be fully optimized. However, we believe that these experiments broadly illustrate which descriptors are responsible for the performance improvement.

Furthermore, we compare performance of the DeepPCM model on all splits with performance on the No-Interaction-Terms model, using both handcrafted and unsupervised-learned descriptors. By comparing these two models, we can investigate the difference between the full model, which is potentially able to model the specific bio-chemical interaction between the protein and the ligand, versus a model that can only recover compound and protein bias in the dataset.

We find that models with interaction terms between the protein and ligand perform significantly better than the No-Interaction-Terms model across all metrics and splits, which is consistent with our expectation that the model is able to make predictions based on specific protein-ligand binding information, and is not merely collapsing to predict the protein or compounds independent propensity for binding, which is what the No-Interaction-Terms-model is restricted to (Table 1).

Just as when using full models with interaction terms, we find that the No-Interaction-Terms model using unsupervised-learned descriptors performs better than the No-Interaction-Terms model using handcrafted descriptors, which suggests that the unsupervised-learned descriptors contain more useful information on which to learn general features of proteins and compounds that make them more or less likely to be binding overall. Additionally, we find that the improvement in performance from using the full models rather than the No-Interaction-Terms models is greater when using unsupervised-learned descriptors than when using handcrafted descriptors. For example, on the temporal split, the unsupervised-learned descriptors improve performance on the DeepPCM model by 16% relative to the No-Interaction-Terms model with unsupervised-learned descriptors,

while the handcrafted descriptors improve performance on the benchmark model by only 6% relative to the No-Interaction-Terms model with handcrafted descriptors (Table 1). This suggests that the unsupervised-learned descriptors facilitate the learning of actual protein-ligand interactions to a greater degree than do handcrafted descriptors.

This finding is supported by the results on the low-protein-coverage split, where Figures 2B and 2C depict the difference between the full models and the No-Interaction-Terms models when using unsupervised-learned descriptors and handcrafted descriptors respectively. The DeepPCM model significantly outperforms the No-Interaction-Terms model when the unsupervised-learned descriptors are used as input ($p = 2.1*10^{-6}$). This is not the case when handcrafted features are used, where the difference in performance between the benchmark model and the No-Interaction-Terms model is not significant ($p = 0.10$).

Taken together, these results indicate that the unsupervised-learned descriptors — despite being more compact representations that were not specifically trained for the PCM task but instead learned in a fully unsupervised manner — are able to not only store more meaningful information about a protein or ligand's overall binding propensity, but also that they are more useful for learning higher-order features required to model the specific binding interaction of a given protein and ligand.

## 5 Conclusion

In this work we proposed a proteochemometric model that utilizes a feature extraction method for both ligand and target that was pre-trained on an unsupervised learning task. We demonstrated how a PCM model based on these descriptors significantly outperformed state-of-the-art hand-crafted descriptors in various experiments, including the impactful low-protein-coverage split. Looking forward, we believe that with sufficient data, PCM models could be powerful enough to replace much of the expensive and time-consuming *in-vitro* experimentation required to develop new drugs, and could also find applications in precision medicine – for example to examine the influence of mutations in viral, cancerous, or otherwise modified proteins on their binding activity. More broadly, we hope that by demonstrating the power of unsupervised pre-trained embeddings on a biological task, this work inspires further research to improve currently existing machine learned representations of compounds and proteins, to generate new representations of biological and chemical entities such as RNAs, antibodies and cell lines, and to apply these representations to solve difficult problems in the life sciences.

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

## A  DEEPPCM MODEL ARCHITECTURE

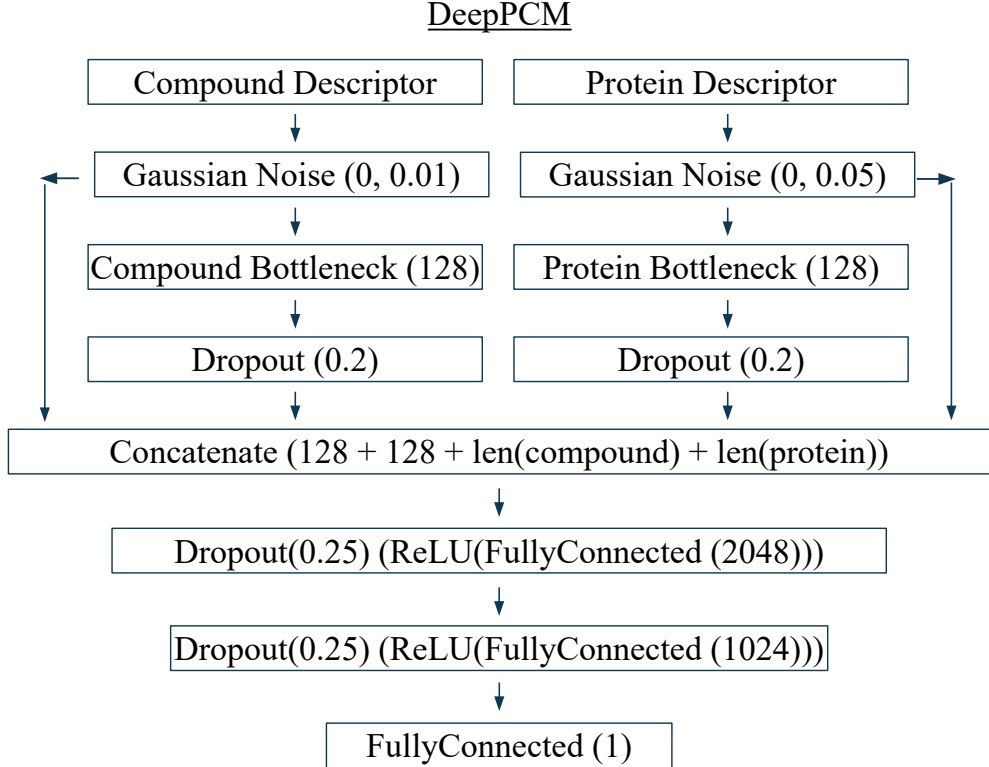

Figure 3: Architecture and Hyperparameters for the DeepPCM model.

## B NO-INTERACTION-TERMS BASELINE ARCHITECTURE

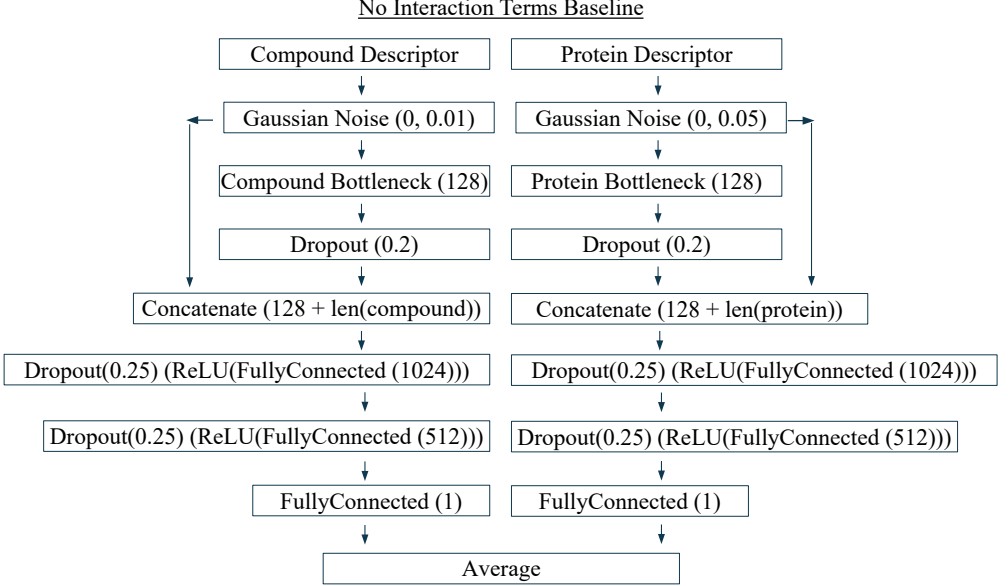

Figure 4: Architecture and Hyperparameters for the No-Interaction-Terms Baseline model.

## C  BENCHMARK MODEL ARCHITECTURE

Benchmark Model

| Compound Descriptor | Protein Descriptor |
| --- | --- |

Concatenate (len(compound) + len(protein))

Dropout(0.25) (ReLU(FullyConnected (4000)))

Dropout(0.25) (ReLU(FullyConnected (2000)))

Dropout(0.25) (ReLU(FullyConnected (1000)))

FullyConnected (1)

Figure 5: Architecture and Hyperparameters for the benchmark model from Lenselink et al. (2017)

## D    CDDD DIAGRAM

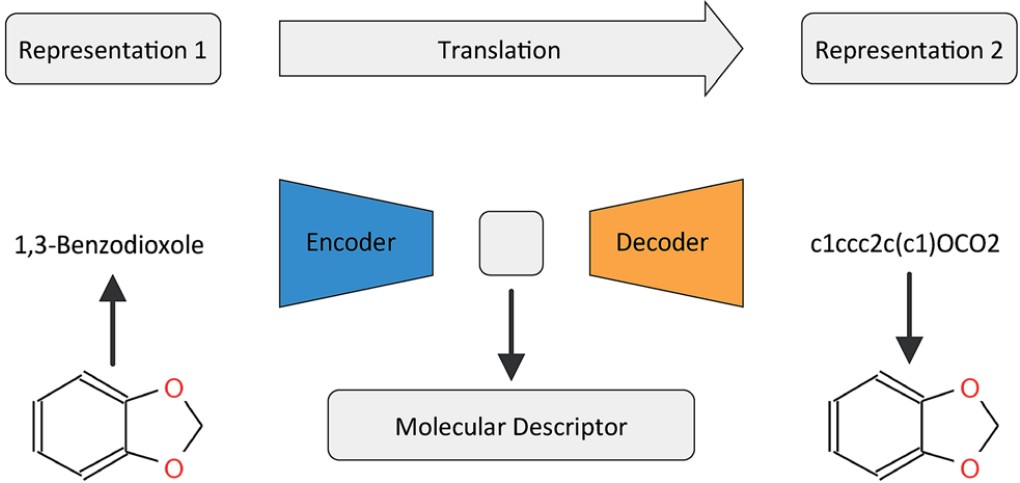

Figure 6: General architecture of the translation model, using the example of translating between IUPAC and SMILES representations of 1,3-benzodioxole. The final CDDD model translates from non-canonical to canonical SMILES representations of compounds. Figure and text taken from Winter et al. (2019) with permission from the authors.

# E UNIREP DIAGRAM

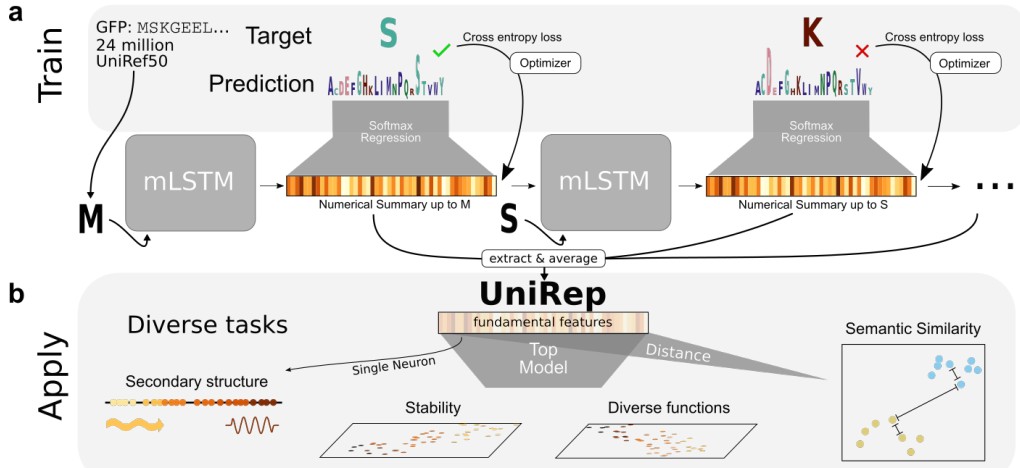

Figure 7: Workflow to learn and apply deep protein representations. **a**. UniRep model was trained on 24 million UniRef50 primary amino acid sequences. The model was trained to perform next amino acid prediction (minimizing cross-entropy loss), and in so doing, was forced to learn how to internally represent proteins. **b**. During application, the trained model is used to generate a single fixed-length vector representation of the input sequence by globally averaging intermediate mLSTM numerical summaries (the hidden states). A top model (e.g. a sparse linear regression or random forest) trained on top of the representation, which acts as a featurization of the input sequence, enables supervised learning on diverse protein informatics tasks. Figure and text taken from Alley et al. (2019) with permission from the authors.

# F  HYPERPARAMETERS

Table 2: The table indicates the hyperparameters that were tested for the DeepPCM model. Bold indicates the hyperparameters that were chosen for the final model based on performance on the validation set of the temporal splits.

| Hyperparameter | Values |
|---|---|
| Learning Rate | 0.05, 0.01, **0.005**, 0.001, 0.0001 |
| Optimizer | SGD, **NAG SGD w/ Momentum**, ADAM |
| Early Stopping Patience | 5, 20, **50**, 100 |
| Compound Input Noise SD | 0, **0.01**, 0.05 |
| Protein Input Noise SD | 0, 0.01, **0.05**, 0.1 |
| Architectures | Wide( >1000 neurons/layer), Deep ( >4 layers) |
| Architecture cont'd | Wide + Deep, Shallow(<3 layers) , **Shallow + Wide** |
| Weight Decay | 0.0001, **0.00001** |
| Learning Rate Decay | **Constant**, On Plateau |
| Dropout | All 0.5, **All 0.25**, All 0.1, 0.5 in early layers and decaying to 0.1 |
| Activation Functions | **ReLU**, TanH, Sigmoid, LeakyReLU, Swish |

