# OpenReview forum: "DeepPCM: Predicting Protein-Ligand Binding using Unsupervised Learned Representations"
_ICLR.cc/2020/Conference — Reject_

### Official Review · AnonReviewer1 · 2019-10-21
**Official Blind Review #1**

**Rating:** 1

**Review:**

This paper proposes to learn representations of protein and molecules for the prediction of protein-ligand binding prediction.

The presentation of this paper is a bit lengthy and repetitive in some cases. The long descriptions of protein/drug descriptors are a nice overview,  but it may be unnecessary as the authors in the end use other works’ embedding.

The author points out that there are interpretability issues & the inability to capture the shape of the substructures with previous ligand descriptors, however, it seems that CDDD also is not interpretable and could not capture the shape as it operates on SMILES strings, although seems to have better predictive performance.

For the protein descriptor, the author is missing several important descriptors that may not have the issues mentioned such as Protein Sequence Composition descriptor.

The technical novelty is very limited. It seems the usage of CDDD and UniRep are its only difference from previous works such as DeepDTA, WideDTA, DeepConv-DTI, PADME and CDDD and UniRep are also from other works. It may be more suitable for a domain journal instead of ICLR which focuses on method innovation.


The experimental setup is solid with realistic considerations. However, it is missing many baselines such as DeepDTA, WideDTA, DeepConv-DTI, PADME and more classic methods such as SimBoost and KronRLS.

**Experience Assessment:**

I have published in this field for several years.

**Review Assessment: Checking Correctness Of Derivations And Theory:**

I did not assess the derivations or theory.

**Review Assessment: Checking Correctness Of Experiments:**

I carefully checked the experiments.

**Review Assessment: Thoroughness In Paper Reading:**

I read the paper thoroughly.

---

### Official Review · AnonReviewer2 · 2019-10-23
**Official Blind Review #2**

**Rating:** 3

**Review:**

This paper tries to solve the protein-legend binding prediction problem in the computational biology field. It uses the learned embedding for protein and legend, separately, from two published papers. Then those two embeddings were inputted to another deep learning model, performing the final prediction. Tested on one dataset, it shows the proposed method can outperform the other baseline methods.

The paper should be rejected for the following reasons:
1. The idea of the paper is not interesting and novel enough. It only used the results from two published papers and then applied another deep learning model on them. The novelty of the paper is limited.
2. The experiment part is not that comprehensive. It indeed performs enough ablation studies. However, all the legendary methods in the computational biology field are not included in the comparison.
3. The experiments are only performed on one dataset. Usually, for application papers, the experiments should be performed against at least 2 datasets to avoid bias.
4. The discussion part is not well-developed. For the paper which focuses on only one task, the more in-depth discussion is expected beyond the simple discussion of the performance. For example, the authors may try to explain the result: why the model used embedding from unsupervised learning is better than the hand-crafted features. Since they shared the same model, the unsupervised embedding should contain more information. Then, what is the additional information?


Some further questions and comments:
1. What's the sequence similarity of the 1226 proteins?
2. Can the model generalize well to a completely new protein?
3. What's the detailed performance of the model on proteins belonging to different families?
4. I guess if the authors check the detailed performance, they will find nonuniform performance across different proteins. I think the authors can also further investigate that.
5. Can the authors train the embedding model as well as the classification model in an end-to-end fashion? This can be more interesting.
6. One big problem of the assay data is that it would not be able to provide the structure information of the interaction between the protein and the legend. Usually, it is a very important piece of information for the biology people. The computational methods based on the assay data will also inherit the flaw.

**Experience Assessment:**

I have published one or two papers in this area.

**Review Assessment: Checking Correctness Of Derivations And Theory:**

I carefully checked the derivations and theory.

**Review Assessment: Checking Correctness Of Experiments:**

I carefully checked the experiments.

**Review Assessment: Thoroughness In Paper Reading:**

I read the paper at least twice and used my best judgement in assessing the paper.

---

### Official Review · AnonReviewer4 · 2019-11-02
**Official Blind Review #4**

**Rating:** 3

**Review:**

The authors present a model with state-of-the-art performance for predicting protein-ligand affinity and provide a thorough set of benchmarks to illustrate the superiority of combining learned low-dimensional embedding representations  of both ligands and proteins. The authors then show that these learned representations are more powerful than handcrafted features such as circular fingerprints, etc. when combined into a model that jointly takes as input both the ligand and protein.

I originally suggested to accept the paper, but agree with the other reviewers that the novelty of the work in this paper likely doesn't meet the bar for acceptance given that the most significant contributions of this paper are around combining good ideas from other papers without much additional novelty.

Major comments

Unless I'm misunderstanding the naming conventions used in Table 1, it seems like an omission not to include the performance of DeepPCM+HP+HC in Table 1 to more fully decouple the performance due to the DeepPCM architecture versus the contribution due to using both unsupervised descriptors in combination.

I might expect the performance of the DeepPCM + HP + HC model (not currently shown unless I'm missing it) to exceed the performance of the NIB + HP + HC model based upon the reasoning given in the discussion (i.e., usefulness of joint input training), even if the individual representations were inferior to the unsupervised counterparts.

Since the primary point of the paper is to illustrate the power of combining the unsupervised representations, I'm surprised the aforementioned performance comparison is not prominent. That is, the performance comparison of DeepPCM+HP+HC versus DeepPCM+UP+UC seems like a very central quantity to present and discuss, but appears to be missing currently.


Minor comments

Table 1 was a bit confusing to me at first, because it appears that UD = UP + UC, and HD = HP + HC, but this wasn't obvious to me initially; I'd change the NIB rows to be NIB + HP + HC and NIB + UP + UC, and then you don't even need to define the UD and HD acronyms, which simplifies the table's cognitive load for me and also makes the relationship between the DeepPCM variants more obvious.

> "On the low-coverage-split, we also find that our method significantly outperforms the benchmark."

It would be good to add the %improvement inline here to parallel the other dataset splits listed just before so that you can directly compare the magnitudes.

> "using unsupervised-learned descriptors than when using handcrafted descriptors"

Would be more compelling if you added the DeepPCM + HC + HP performance to Table 1.

> "All hyperparameter optimization of our model was performed on the temporal split"

I think it would improve the paper if the extent to which various architecture choices were optimized over could be included as well; for example, which types, layer sizes, and depths of network architectures were considered in the hyperparam tuning, and any accompanying justification for these design choices.


**Experience Assessment:**

I have published in this field for several years.

**Review Assessment: Checking Correctness Of Derivations And Theory:**

N/A

**Review Assessment: Checking Correctness Of Experiments:**

I carefully checked the experiments.

**Review Assessment: Thoroughness In Paper Reading:**

I read the paper thoroughly.

---

### Public Comment · ~Christian_Dallago1 · 2019-10-15
**Nice work, needs some clarifications**

Summary:
This work describes a novel way to combine data-driven representations for small molecules as well as proteins for the prediction of protein-ligand binding. The authors show that these novel representations, which are learned in an unsupervised fashion, improve over traditionally crafted features.


# Things that we appreciated:
+ Network graphs: clear and easy to follow

+ Approach: novel and interesting, laying out well the disadvantages of traditional chemical/protein encoding

# A few notes:
- The authors claim that there might be similar data items in the random train and test splits (as similar chemicals or proteins might not be accounted for randomly). However, in the temporal splits there is no way of telling that there is no similarity between a compound being analyzed in the past w.r.t. more recently, leading us to believe that temporal splits address only partly the bias between training and testing

- Validation set for temporal split: it's not clear how the authors split the temporal splits for validation, training and testing. Especially, as this split will very likely have the same bias as the authors describe for the random split

- It is not clear how the authors try to calculate the standard error and how they apply bootstrapping. This might be due to unclear wording or lack of details

- Low coverage splits: why wasn't a traditional approach (e.g. homology reduction) used for this?

- Table 1: the results on the low coverage splits are missing, making it difficult to easily compare across different approaches

- Writing: Table 1 should appear in the results instead of the methods, and the discussion should not introduce new methods (e.g. evaluation of the individual impact of each descriptor type, in the second paragraph of discussion)

# Suggestions:
* UniRep was used for the embedding of proteins, but another method is available (SeqVec). We suggest also trying this encoder and see if there is any improvement over using UniRep. We acknowledge a conflict of interest, as SecVec was produced by our lab (but in another ICLR open review publication (https://openreview.net/forum?id=Skx73lBFDS), SeqVec seemed to spawn better results then UniRep for the problem of GO annotation prediction)

* For the test sets: create three different test sets based on difficulty / novelty of the proteins, e.g.  based on similar to the different test sets evaluated on protein-protein interaction prediction information (Evolutionary profiles improve protein–protein interaction prediction from sequence, Hamp&Rost, 2015)

This comment was co-authored by members of RostLab: Michael Heinzinger, Tobias Olenyi, Oscar Llorian, Maria Littmann, Christian Dallago

---

> ### Author Response · Authors · 2019-10-17
> **Thank you for the helpful comments! (1)**
>
> We thank the commenters for their input and for their helpful suggestions for further directions to improve the paper. We address the specific comments below.
>
> - The authors claim that there might be similar data items in the random train and test splits (as similar chemicals or proteins might not be accounted for randomly). However, in the temporal splits there is no way of telling that there is no similarity between a compound being analyzed in the past w.r.t. more recently, leading us to believe that temporal splits address only partly the bias between training and testing
>
> While the commenters are correct that the temporal split may not fully address the bias between training and testing, we include the temporal split in the results for two reasons
>      1. The temporal split creates a more realistic test set, since in real-world usage of PCM models one would train on all currently available data. This split simulates this case.
>      2. The temporal split was used by Lenselink et al to benchmark their handcrafted-descriptor models – for the sake of comparability we stick to the same evaluation schema.
> On revision, we will include these details – in particular to state that the temporal split may not fully address the bias between train and test, but that we include it because it is a realistic split.
> We note that we are aware of more rigorous splitting schemes based on protein sequence similarity, and of the contribution of Rost & Sander to these ideas in their seminal work from 1993 (https://www.ncbi.nlm.nih.gov/pubmed/8356056)
>
>
> - Validation set for temporal split: it's not clear how the authors split the temporal splits for validation, training and testing. Especially, as this split will very likely have the same bias as the authors describe for the random split
>
> We apologize for not clearly explaining the construction of the validation set for the temporal split. We used the same split procedure as Lenselink et al, where the test set contains data generated from assays done in 2013 and later, while the training set and validation set is constructed by randomly splitting the pre-2013 data. By ensuring both training and validation is done on pre-2013 data, we seek to closely mimic the real-world case of training on all currently available bioactivity data, and to avoid the bias from random splitting as much as possible.  We will make this clearer in revision.
>
>
> - It is not clear how the authors try to calculate the standard error and how they apply bootstrapping. This might be due to unclear wording or lack of details
>
> We apologize for our unclear writing in this section. For the random split, each bootstrap set is constructed by separating a random test set, then separating a random validation set from the remaining data, then resampling with replacement from the training set to create our bootstrap training set. For the temporal split, the test set is fixed, containing all bioactivity data generated from assays done in 2013 and later. Then, in each set we randomly partition the pre-2013 data into a training and validation set, then resample with replacement from the training set to create our bootstrap training set. In all cases, we also save our random seeds for future reproducibility. The standard error we report is then computed from the standard deviation of the performance of the bootstrap samples. In revision we will make this section more clear and include the details in the main body or in the appendix.
>
>
> - Low coverage splits: why wasn't a traditional approach (e.g. homology reduction) used for this?
>
> We ask the commenter to elaborate what they are referring to as homology reduction, other than the mathematical concept in algebraic topology. Do they mean some technique involving construction of a hold-out set where the hold-out set comprises a distinct family of proteins excluded from the training set?
>
>
> - Table 1: the results on the low coverage splits are missing, making it difficult to easily compare across different approaches
>
> The low-coverage split does not lend itself to computing standard errors in the same manner as the temporal or random splits, since we already create several splits of the data – based on which five proteins have been assigned to the test set. Also, in this case, we find the average performance is not as useful as visualizing the difference in performance with respect to each protein.

---

> > ### Author Response · Authors · 2019-10-17
> > **Thank you for you helpful comments (2)**
> >
> > - Writing: Table 1 should appear in the results instead of the methods, and the discussion should not introduce new methods (e.g. evaluation of the individual impact of each descriptor type, in the second paragraph of discussion)
> >
> > We were not sure where to put the evaluation of the individual impact of each descriptor type, since it is a method but is more for investigative purposes rather than a key point of our results. It seemed redundant to explain it in both the methods and then in the discussion. We will make changes in the revision to improve the flow while ensuring the discussion does not introduce new methods.
> >
> >
> > * UniRep was used for the embedding of proteins, but another method is available (SeqVec). We suggest also trying this encoder and see if there is any improvement over using UniRep. We acknowledge a conflict of interest, as SecVec was produced by our lab (but in another ICLR open review publication (paper) SeqVec seemed to spawn better results then UniRep for the problem of GO annotation prediction)
> >
> > Thank you for this suggestion. We were unaware of the SeqVec – there seem to be several new protein embeddings coming out in the past year, and we felt that benchmarking across all these different embeddings would be out of the scope of this paper. We will evaluate on SeqVec and if the results on SeqVec are better, we will use these results for the paper instead of UniRep.
> >
> >
> > * For the test sets: create three different test sets based on difficulty / novelty of the proteins, e.g. based on similar to the different test sets evaluated on protein-protein interaction prediction information (Evolutionary profiles improve protein–protein interaction prediction from sequence, Hamp&Rost, 2015)
> >
> > We thank the commenter for this suggestion – these types of test sets would make the evaluation much stronger. We will explore the addition of more strictly designed test-sets as demonstrated in the paper.

---

### Decision · Program_Chairs · 2019-12-19

**Decision:**

Reject

**Comment:**

This paper uses unsupervised learning to create useful representations to improve the performance of models in predicting protein-ligand binding. After reviewers had time to consider each other's comments, there was consensus that the current work is too lacking in novelty on the modeling side to warrant publication in ICLR. Additionally, current experiments are lacking comparisons with important baselines. The work in its current form may be better suited for a domain journal.